# Forest Climax Phenomenon: An Invariance of Scale

**Raimundas Petrokas**

Institute of Forestry, Lithuanian Research Centre for Agriculture and Forestry, LT-53101 Kaunas distr., Lithuania; raimundas.petrokas@lammc.lt

**Abstract:** We can think of forests as multiscale multispecies networks, constantly evolving toward a climax or potential natural community—the successional process-pattern of natural regeneration that exhibits sensitivity to initial conditions. This is why I look into forest succession in light of the Red Queen hypothesis and focus on the key aspects of ecological self-organisation: dynamical criticality, evolvability and intransitivity. The idea of the review is that forest climax should be associated with habitat dynamics driven by a large continuum of ecologically equivalent time scales, so that the same ecological conclusions could be drawn statistically from any scale. A synthesis of the literature is undertaken in order to (1) present the framework for assessing habitat dynamics and (2) present the types of successional trajectories based on tree regeneration mode in forest gaps. In general, there are four types of successional trajectories within the process-pattern of forest regeneration that exhibits sensitivity to initial conditions: advance reproduction specialists, advance reproduction generalists, early reproduction generalists and early reproduction specialists. A successional trajectory is an expression of a fractal connectivity among certain patterns of natural regeneration in the multiscale multispecies networks of landscape habitats. Theoretically, the organically derived measures of pattern diversity, integrity and complexity, determined by the rates of recruitment, growth and mortality of forest tree species, are the means to test the efficacy of specific interventions to avert the disturbance-related decline in forest regeneration. That is of relevance to the emerging field of biocomplexity research.

**Keywords:** habitat; regeneration; succession; dynamics

---

## 1. Background

New forestry practices combine technical efficiency, economic and environmental performance, resistance and resilience to disturbances; however, forests demonstrate a greater vulnerability to anthropogenic impacts and gradual climatic changes—the two major forms of disturbance occurring today—than to large infrequent disturbances [1,2]. "Current views of succession emphasise ongoing process rather than the climax community as the stable end point or product" [3], and "the sharp distinction between successional and climax forests is widely applied today, with major implications for conservation practices and land-use policy" [4]. Forests are intensely harvested for timber and biofuel and are never allowed to recover the natural climax state; the water regulation potential of such forests is low, while their susceptibility to fires and pests is high [5]. Therefore, we have to ask here: Are not forest resistance and resilience to disturbances the result of ecological invariance? Because the essence of ecological patterns and processes is invariance [6]. Ecological invariants are known as "scaling" and "power laws" that describe power relationships across species or across ecological systems [7]. "Scale invariance, also called scaling, or scale-free dynamics, implies that the phenomenical or phenomenological dynamics are driven by a large continuum of equally important time scales, rather than by a small number of characteristic scales" [8,9]. "By scale invariance in ecology, we mean that scales are ecologically equivalent so that the same ecological conclusions may be drawn

from any scale statistically" [10]. A characteristic indicator of scale invariance is when the frequency distribution of the events of self-organisation decays only as a power law, reflecting the self-similarity of the critical state [11–13]. For instance, Manrubia and Solé [14] performed an extensive study of a real rainforest in Barro Colorado Island, Panama, and found strong evidence of a self-organised critical state in the power laws followed by the magnitudes of the system, both in space (fractality, correlation function, clearings and tree size distributions) and time (biomass fluctuations). In general, common probability patterns arise from simple invariances; invariance defines scaling relations and probability patterns [15]. Nottale [16] offers fractal space–time as a method for establishing an invariance of scale. What is more, the fundamental principle underlying the theory of invariance is that the laws of nature always have the same form for all observers [17]. This leads to a reconsideration of the traditional approach to forests focused on long-term dynamics in favour of a successional approach [18–22], which has emerged from the recognition that "even intense natural disturbances leave biological legacies and spatial heterogeneity in the new forest, which contrasts with the simple and homogeneous environment that is often the outcome of traditional harvesting practices, particularly clear-cutting" [23].

The goals of the review were to (1) present the framework for assessing habitat dynamics and (2) present the types of successional trajectories based on tree regeneration mode in forest gaps. A synthesis of the literature was undertaken within the context of the fractal fragmentation and connectivity of landscape habitats. "Fractals are dynamic process-structures that etch time into space; by illuminating fractals, we self-reflexively illuminate the observer in the observed in nothing short of nature herself" [24]. However, it is necessary to clarify that, contrary to ideal fractals, landscape patterns can be considered fractals only for limited scale intervals; this is due to the dimensionality of successional elements, determined by their regeneration response to disturbance [25]. Watt [26] was the first to link space and time at the landscape scale. There are two main models of forest dynamics, developed from Watt's seminal idea of patch dynamics, i.e., the patch–mosaic model and the gap–phase model [27]. The patch hierarchy approach, based on the hierarchical patch dynamics paradigm [28] that integrates hierarchy theory with the patch dynamics perspective, has proven useful in scaling landscape patterns and processes [29]. Scale is a main concept in landscape ecology that focuses on the influence exerted by spatio-temporal patterns on the organisation of, and interaction among, functionally integrated multispecies ecosystems [30].

## 2. Fractal Forest

In mathematics, symmetries have the peculiar status of being both invariant and invariant-preserving transformations, which is why a fractal, being a highly nontrivial representation of the two fundamental symmetries of nature, dilation ($r \rightarrow ar$) and translation ($r \rightarrow r + b$), exhibits self-similarity or pattern integrity—the retention of copies of itself on a hierarchy of scales [19,31–35]. In other words, a fractal is known as expanding symmetry or evolving symmetry [36]. By virtue of occupying the exact portion of the geometrical space that it occupies, a fractal has a non-integer dimension that is less than, or equal to, the Euclidean dimension of the space it occupies [37,38]. If a fractal space in which a dynamic process takes place becomes a Euclidean space with integer dimension, this means that the process has left its strange attractor (i.e., an attractor of fractal dimension) and tends toward, or already is in, the state with a lower number of possible directions of further evolution [39]. For instance, Palmer [22] demonstrated that increasing fractal dimension (decreasing spatial dependence in a landscape) allowed more species to exist per microsite and per landscape. It must be noted, nevertheless, that, when dealing with fractals, the fundamental characteristic of being differentiable is missing. Therefore, it is a challenging problem to define operators on fractal sets [40]. "Strict fractal objects require infinite power–law scaling, which fails to address the limited range of scale invariance observed in nature" [41]. So, in the end, the question is whether an ecological system manifests a physically meaningful degree of fractal connectivity between its subunits [10,42,43].

With the discovery that a set of symbols has been used by nature to encode the information for the construction and maintenance of all living things, fractals—the invariant sets of chaotic systems—serve

paradoxical functions as physical boundary keepers, both to separate and connect various subsystems and levels of being [24,44–46]. The key notion of a fractal is that it possesses structures on a hierarchy of scales generated by the reiteration of a mathematical formula [35], which is "a form of feedback, where the answer to the formula recycles into the original formula to generate the next solution" [47]. Therefore, the complexity of dynamic process-structures is measured by their fractal dimension [24,48]. For instance, the number $N(x)$ of objects with a characteristic linear dimension greater than $x$ can be given by $N(x) \sim x^{-D}$, where $N(x)$ is a number measure corresponding to the scale unit $x$ and $D$ is the fractal dimension [8]. The value of the scaling exponent of the number–size relationship may vary widely, and the power–law scaling only holds over a finite range of time scales in real landscapes [49,50]. A power–law distribution of the probability density function of the pieces of an object in space or the parts of a process in time is evidenced in a straight line on a plot of log (number) vs. log (size) [51]. To estimate the habitat patch fractals, an alternative power–law distribution can be written: $N(m) = Cm^{-b}$, where $N(m)$ is the number of habitat patches with a biomass greater than $m$, $C$ is a constant and $b$ is a scaling exponent; noting that $m \sim x^{-3}$, we can find from a comparison with the number–size fractal distribution that $D = 3b$ [8]. From a forest management perspective, the patchwork of habitats in varying successional stages of recovery may correspond to forest stand development patterns, and the biomass to the stand volume. The power–law distribution of fractal fragments can be used as an indicator of the fragmentation of the landscape habitat into patches and landscape connectivity change [8]. Fractal fragmentation is often a scale-invariant process, but nevertheless most ecological patterns and processes show scaling thresholds at which abrupt changes in scaling relationships occur, corresponding to shifts in underlying mechanisms [52].

## 3. Coexistence

The basic components of a network of elements whose creation, evolution, destruction and interaction cause the emergence of a particular behaviour or feature that cannot be reduced to the properties of an individual system's components are called coexisting attractors [32,53,54]. An attractor—a region in state space that a system can enter but not leave—is a mathematical model of causal closure [55]. Closure usually results from the nonlinear, feedback nature of interactions [46,56]. We can generalise an attractor as any state toward which a dynamical system tends to evolve. Within the framework of chaos theory, the overloaded vegetation climax is considered a "strange attractor": the smallest invariant set of the events of self-organisation that exhibit exponential sensitivity to initial conditions [13,57–61]—"sensitivity in fact to any numerical rounding at any calculational step, not necessarily at the initial time" [62]. According to Anand [63], "this attractor itself is moving on a deterministic path imposed on the process by a highest order environmental constraint, the long term evolution of the Earth's climate". More than that, the final state toward which an ecosystem tends to evolve usually depends on the initial conditions involving several coexisting attractors [64–69]. As Allesina and Levine [70] put it, "just a handful of limiting factors can generate the coexistence of many species, a feature of intransitive networks". Climax forest is an excellent example of such an ecosystem in which climate, landscape, vegetation and fauna are closely interconnected: when one of these components is destroyed, whether partially or completely, the other components undergo an equally violent change [71]. Therefore, it follows that forest climax should be associated with habitat dynamics driven by a large continuum of ecologically equivalent time scales: the frequency of occurrence of an event of a given magnitude $x$ is inversely proportional to some power $\alpha$ of its magnitude, $f(x) \sim x^{-\alpha}$ [33,72]. $f(x) \sim x^{-1}$—a critical dependence—is often associated with a feedback dynamic that creates a stable equilibrium at a critical point. It provides a general mechanism for the emergence of scale-free networks with the power–law degree distribution [16,73–78]. However, real-world networks do not follow power–law degree distribution over the whole range of a degree. In many real-world networks, scale-free property coexists with a hierarchy of nodes, low node separation and high clustering [79]. The hierarchy always follows a pair of exponential laws and a power law; it appears if a certain pattern is added at each time unit into the network [79,80]. "Networks which

display scale-free properties are the most hierarchical" [81]. Hierarchical network structure promotes a dynamical robustness, the origins of which are in the understanding of the impact of node failures on the integrity of a network [82]. This is important for many disciplines, as many real-world networks are organised into many small, highly connected modules that combine in a hierarchical manner into larger units, with their number and degree of clustering following a power law [83,84]. Power–law distributions can be produced by endogenous processes like feedback loops, self-organisation, network effects, etc., so the key problem is to understand why nature gives rise to the wide diversity of degree structures found in real-world networks and why scale-free networks are rare [10,85].

We can think of forests as multiscale multispecies networks, constantly evolving toward a climax or potential natural community—the successional process-pattern of natural regeneration [3,12,74,86–92]. Known to ecologists as secondary forest succession, natural regeneration is the regrowth and reestablishment of the forests, recovering from natural or human disturbance [4,91–94]. Natural regeneration, however, is not achieved or accomplished—it is lived and evolved—and this is especially true when there is a focus on ecological self-organisation: organisms connected in communities transform the ecosystem while transforming themselves, and the chemical outputs of organisms, self-produced through feedback loops, are used by other organisms to facilitate their own self-reproduction [95,96]. To put it simply, the biology of self-replication is self-referential, as embodied by nucleic acid replication mechanisms; self-reference is "the hinge upon which levels of serial inclusiveness intercross" [24]—a critical scale of a phenomenon [12,47,97–101]. Furthermore, that is why I look into forest succession in light of the Red Queen hypothesis: life has evolved in order to stay extant, or else go extinct. The Red Queen hypothesis, as formulated by van Valen [102], is similar to that of a system obeying a self-organised criticality, which means that a given Red Queen phenomenon is caused by the system that organises its critical state by itself [103,104].

Within biology, the developmental process of organisms as well as their metabolisms, growth and learning have been identified as self-organising processes; nevertheless, there is yet no unique theory of self-organisation [105]. One of the objectives of the present article was, therefore, to give prominence to the key aspects of ecological self-organisation: dynamical criticality, evolvability and intransitivity. "Dynamical criticality, a central property for the functioning of a living organism, naturally emerges as a consequence of evolution that favours evolvability" [106]. Dynamical criticality explains evolution by reference to the broad internal disposition of a population to evolve in order to stay extant, rather than any actual evolutionary trajectory of populations by capturing the influence that the internal features of populations can have upon the outcomes of evolution [107]. Taking into account the above, I propose the conceptual framework for assessing habitat dynamics at a network–system–trajectory interface (Table 1): The network, because nature can be viewed in terms of multilevel, multidimensional hierarchies of inter-related event clusters that form a metaheterarchy, or a heterogeneous general hierarchy [24]; the system, because the experience of events can be viewed as a summary of the facts through which the events took place—a fact pattern [108,109]; and the trajectory—an expression of a relation among certain fact patterns in the network [79,110].

**Table 1.** The conceptual framework for assessing habitat dynamics.

| Aspects | Concepts | Verifiers | |
| --- | --- | --- | --- |
| | | Pattern | Process |
| Intransitivity | Network | Diversity [70,111] | Robustness [82,83,112,113] |
| Criticality | System | Integrity [114] | Fitness [115–121] |
| Evolvability | Trajectory | Complexity [22,107] | Inclusiveness [24,101,122] |

## 4. Successional Species Turnover

The view of succession was developed from the patchwork pattern of habitats in varying stages of recovery from human disturbances. However, a compositional shift, in which post-disturbance stands dominated by fast growing shade-intolerant tree species are eventually replaced by late seral, shade-tolerant species, is not a simple unidirectional sequence of stages, but rather a complex model subject to differential species responses to factors such as physical site conditions, initial stand composition and intermediate disturbance effects [123]. Furthermore, "although successional stages are defined by characteristics of a forest stand, successional trajectories are fundamentally determined by rates of recruitment, growth, and mortality of populations of the component tree species" [124]. "Trees define the communities that they inhabit, are host to a multiplicity of other organisms and can determine the ecological dynamics of other plants and animals" [114]. In woodland habitat modelling, tree species form the basis of the parameters used to represent a range of component species within a particular process-pattern of the events of self-organisation and to derive a network. Moreover, unlike the action of seasons and natural disasters, long-term change in the composition of communities is brought about by the activities of living organisms which themselves inhabit the environment [19]: "Many of our rarest species are associated with ancient trees and only occur where there has been a continuous cover of old trees back through time on the site" [125]. For these reasons, the use of measures that account for the Red Queen dynamics of interacting populations of the component species that form the continuous cover of trees through time on the site may provide new ways to monitor succession and test the efficacy of specific interventions to modify the disturbance-related changes in successional process properties: robustness, fitness and inclusiveness (Table 1) [21,120,122,126–132]. "It has been suggested that Red Queen dynamics underlie a large number of important biological processes, some of which are still poorly understood, such as genetic recombination and sexual reproduction" [120].

The shift in dominance by shade-intolerant tree species to shade-tolerant tree species is the most generalisable and predictable feature of successional pathways [4]. The most distinctive difference between shade-intolerant and shade-tolerant tree species is that the former are incapable of establishing themselves in a forest understorey (Table 2), because their seeds do not accumulate in a long-lived seed pool. Instead, they germinate immediately upon dispersal or soon thereafter [133]. Shade-intolerant tree species, colonising from a refuge site, fast-growing, having low wood density, branching with axial differentiation, short-lived, rapidly establishing on disturbed sites, showing early reproduction, high fecundity and large dispersal, often bear numerous small seeds, annually produced and wind- or animal-dispersed [4,134–137]. Shade-tolerant tree species, advance reproduction-dependent, slow growing, branching without axial differentiation, long-lived, gradually replacing intolerants in the absence of disturbance, often bear few seeds, larger in size, sometimes masting, sometimes dispersed only locally and by diverse dispersal agents. Still, in the context of forest dynamics, the low light survival/high light growth tradeoff is "only one of the many possible strategies for trees to differentiate along a disturbance gradient", and "it is unlikely to function as an important mechanism for the stable coexistence of several tree species" [134]. This is not to deny the importance of the tradeoff in determining the successional status of species, yet "successional species turnover, in which pioneer species are being replaced by shade-tolerant species, already starts at the very early years of succession" [138]. The model of initial floristic composition postulates that most late successional species (like shrubs and trees) that will later dominate the community are already available at the onset of succession: "They are either part of the soil seed bank or present with vegetative propagules, rhizomes, or a sapling bank" [116]. So, "in actual practice, the distinction between a 'successional' and a 'climax' forest is subjective; there is no magical moment when a forest stops undergoing succession" [4].

**Table 2.** The types of successional trajectories based on tree regeneration mode in forest gaps. Chazdon's et al. [124] successional trajectories (Clark and Clark's [139] species groups A–D) correspond roughly to Whitmore's [140] species groups 1–4, having increasing "pioneer index", Yamamoto's [141] four major types of tree regeneration mode in gaps (numbered I–IV) and Petrere's et al. [90] four community types. Modified from Franklin [142].

| Growth | Establishment | |
|---|---|---|
| | **Forest** | **Gaps** |
| Forest | Old-growth specialists (A) | Successional generalists (C) |
| | Establish and grow in dark forest; shade-tolerant species. Low potential and average growth rates, especially as juveniles. (1) | Establish in gaps, grow best in gaps, but can survive as saplings in closed forest. Higher juvenile growth potential than groups A or B. (3) |
| | Advance regeneration/gap filler/understorey tree (III) | Gap coloniser/gap filler/canopy tree/gap maker (II) |
| Gaps | Old-growth generalists (B) | Successional specialists (D) |
| | Establish in shade but show increased association with gaps as saplings. Growth rates as low as group A as juveniles, increasing with size. (2) | Establish and grow best in gaps at all juvenile stages; shade-intolerant species. Highest growth potential, especially as juveniles. (4) |
| | Advance regeneration/gap filler/canopy tree/gap maker (I) | Gap coloniser/canopy tree/gap maker (IV) |

## 5. Conclusions

Is the Red Queen hypothesis similar to that of the system obeying a power–law sensitivity to initial conditions? Well, yes it is—and that is all about the time scale of self-reference. All climax communities exhibit sensitivity to initial conditions at any time scale of self-reference. There are four types of successional trajectories within the process-pattern of forest regeneration which exhibits sensitivity to initial conditions: advance reproduction specialists—gap fillers/understorey trees—establishing and growing in a dark forest; advance reproduction generalists—gap fillers/canopy trees/gap makers—establishing in the shade but showing increased association with gaps as saplings; early reproduction generalists—gap colonisers/gap fillers/canopy trees/gap makers—establishing in gaps, growing best in gaps, but surviving as saplings in a closed forest; early reproduction specialists—gap colonisers/canopy trees/gap makers—establishing and growing best in gaps at all juvenile stages. Recognising that dynamical criticality, a central property for the evolvability and intransitivity of living organisms, naturally emerges as a consequence of ecological self-organisation, forest climax should be associated with habitat dynamics driven by a large continuum of observer-invariant time scales. For this reason, the organically derived measures of pattern diversity, integrity and complexity, determined by rates of recruitment, growth and mortality of forest tree species that form a climax or potential natural community, are the means to test the efficacy of specific interventions to modify the disturbance-related changes in successional process properties: robustness, fitness and inclusiveness.

**Funding:** This research did not receive any specific grant from funding agencies in the public, commercial or not-for-profit sectors.

**Acknowledgments:** I am especially grateful to David Smart, who, as an English native speaker, willingly assessed the text of my review. This study was supported by the long-term research programme "Sustainable forestry and global changes", implemented by the Lithuanian Research Centre for Agriculture and Forestry.

**Conflicts of Interest:** The author declares no conflicts of interest.

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
