# Peer review of "Forest Climax Phenomenon: An Invariance of Scale"

_forests, doi:10.3390/f11010056_

Round 1

Reviewer 1 Report

Review of:  Forest Climax Phenomenon: An Invariance of Scale

For: Forests

December 2019

Summary

This manuscript describes forests from the perspective of fractals and leads into a theoretical model of types of tree species based on their successional status.

Strengths

This manuscript offers a good theoretical model for thinking about successional trees from a neutral theory perspective (species as replaceable categories).

Weaknesses

The biggest concern is the leap from the mathematical theory to the theoretical model with not much connecting them.  There needs to be some specific examples and data from the literature to support the categorization of the tree species.  I am also concerned about what appears to be extensive quotations throughout the manuscript.

Specific Suggestions

In my experience, quotations are only used in Science when something specific needs to be shared. It is extremely rare.  This manuscript seems to be largely based on quotes (assuming that all of the single quoted sentences are directly from sources).  I recommend putting the ideas into your own words while citing the sources of the ideas. The emphasis on climax forests is disturbing given that Clements’ ideas about the climax community were challenged by Gleason, and Whittaker’s data supported Gleason’s individualistic concept of plant communities. I believe the author needs to recognize this progression in ecological thinking.  I am troubled by the quick leaps from fractal theory to habitat dynamics to successional types. The conclusion about successional types could stand by itself without the fractal theory.  That said, to reach the successional types conclusion, the author needs to build in support for the categories.  Specific examples from multiple forest types would help.  It would be even better to synthesize data from different studies that provide examples of these successional types.

Recommendation

Given the heuristic value of the successional type model in this manuscript, I recommend that the manuscript need rather large changes.

Reviewer 2 Report

The manuscript contains significant research in forest ecology. The information of paper is valuable for new knowledge.
Author of the paper presents new facts and research results.
The materials are selected sufficiently correctly and correspond to the aim set out in the paper.
Results are clearly presented.
The amount of presented materials are sufficient.
The author has logically interpreted the results of the carried out research.
The conclusions arise from the obtained results and their interpretation.
References to literature appropriate to the subject matter of the paper.

Round 2

Reviewer 1 Report

The authors have made improvements to the manuscript.